# What cancer research makes the news? A quantitative analysis of online news stories that mention cancer studies

**Laura Moorhead**[1]*, **Melinda Krakow**[2], **Lauren Maggio**[3]

**1** Department of Journalism, San Francisco State University, San Francisco, California, United States of America, **2** John D. Bower School of Population Health, University of Mississippi Medical Center, Jackson, Mississippi, United States of America, **3** Uniformed Services University of the Health Sciences, Bethesda, Maryland, United States of America

* lauralm@sfsu.edu

**Data Availability Statement:** To facilitate transparency and replication of our methods, we have made our computer code and the project's

## Abstract

Journalists' health and science reporting aid the public's direct access to research through the inclusion of hyperlinks leading to original studies in peer-reviewed journals. While this effort supports the US-government mandate that research be made widely available, little is known about what research journalists share with the public. This cross-sectional exploratory study characterises US-government-funded research on cancer that appeared most frequently in news coverage and how that coverage varied by cancer type, disease incidence and mortality rates. The subject of analysis was 11436 research articles (published in 2016) on cancer funded by the US government and 642 news stories mentioning at least one of these articles. Based on Altmetric data, researchers identified articles via PubMed and characterised each based on the news media attention received online. Only 1.88% (n = 213) of research articles mentioning US government-funded cancer research included at least one mention in an online news publication. This is in contrast to previous research that found 16.8% (n = 1925) of articles received mention by online mass media publications. Of the 13 most common cancers in the US, 12 were the subject of at least one news mention; only urinary and bladder cancer received no mention. Traditional news sources included significantly more mentions of research on common cancers than digital native news sources. However, a general discrepancy exists between cancers prominent in news sources and those with the highest mortality rate. For instance, lung cancer accounted for the most deaths annually, while melanoma led to 56% less annual deaths; however, journalists cited research regarding these cancers nearly equally. Additionally, breast cancer received the greatest coverage per estimated annual death, while pancreatic cancer received the least coverage per death. Findings demonstrated a continued misalignment between prevalent cancers and cancers mentioned in online news media. Additionally, cancer control and prevention received less coverage from journalists than other cancer continuum stages, highlighting a continued underrepresentation of prevention-focused research. Results revealed a need for further scholarship regarding the role of journalists in research dissemination.

complete data set publicly accessible at https://zenodo.org/record/4448259#.YCQ32NhKg2w.

**Funding:** The author(s) received no specific funding for this work.

**Competing interests:** The authors have declared that no competing interests exist.

## Introduction

The news media are a crucial source of public information that can spur people into action and shape their health beliefs and behaviors [1–6]. Journalists reporting on health and science often include hyperlinks leading to original studies in peer-reviewed journals. These studies are typically funded by the US government, the world's largest financer of cancer research [7], and come with a mandate that research be made publicly available [8,9].

The relatively new phenomenon of direct public access to research [10,11] is relevant to understanding the characteristics of journal articles that are most covered by journalists [1]. It influences how the public, including government officials, policy-makers, health communications professionals, and physicians [12–14], access and understand research about cancer [15–17], the most reported-on disease in US news [18]. Yet, little is known about the alignment between the characteristics of US federally funded research articles and the ways journalists report on them.

Research has generally focused on health news coverage [19–21] in relation to what information people seek [22,23], disease mongering [24–26], scientific uncertainty [26,27] and exaggeration [26–29] or inaccuracy [24] of information. Typically, studies include either a broad mix of mass media (e.g., press releases, newspapers, magazines, radio, television, and the Internet) [30] or a narrow focus on legacy media, typically top-tier traditional print news publications with a national online presence (e.g., *Los Angeles Times*, *New York Times*, *Washington Post*) [31]. Neither media type adequately reflects the growing landscape of digital-native news organizations (e.g., Breitbart News Network, Buzzfeed, Politico, Vox) that publish only online and remain largely outside the attention of researchers [3,27,32,33]. These news publications push the boundaries of conventional journalism [3] and are part of an emerging newscape in which content is shared exponentially on the Internet and across social media platforms. When researchers have considered these news publications, they have typically been framed as secondary to traditional legacy outlets in terms of importance and quality [34–36]. However, there is a need for scholarship both "to capture the diversity" of the field and to recognize the diversity of news and audiences found online.

Audiences for digital-native publications differ from those of legacy media [37,38]. Pew found that 28% of the public typically relied on non-traditional news providers [38]. Notably, younger people increasingly viewed digital-native news rather than legacy media [39]. The further growth of digital-native news has been spurred by the public's increasing distrust of the news media [40]. Additionally, audiences of digital-native and local publications, in particular, tended to be from communities in which cancer mortality remained disproportionately high [41]. These audiences were often less wealthy and educated [42], as well as older and with a higher proportion of people of color [41].

From a public health perspective, reaching these audiences with understandable and accurate medical science is crucial. There is a need to examine journalistic digital-native news and popular online niche and local news publications independent of other forms of mass media and alongside more traditional news publications. For this paper, "journalistic" media is defined as the activity of professional reporters and editors working for news publications to gather, assess and present news and information to help people become more knowledgeable and able to make better decisions [43]. This stands in contrast to mass media, which also include non-profit and governmental news organizations, news aggregators and press release news wires among other content providers. (As discussed under Methods, this paper relied on Pew Research Center's 2016 State of the Media report to define a publication as journalistic).

Journalistic news stories are among the most common ways people access health information [44,45]. Through their news choices and online clicks, people largely control how they

attain information, including scientific research. Among US adults, 37% say they have used news stories with scientific information to help them make decisions about everyday life [45]. Yet, journalists act as gatekeepers, finding, selecting and framing what health information is mentioned and then amplified in news stories [46]. The role of gatekeeper is double-edged and not free from influence (e.g., journal venue, press releases, authors' institutional status and scientific impact) [28,46,47]. Journalists can highlight and omit important health research that is relevant to the public. They also make choices that can privilege a type of research. For instance, researchers reported that journalists generally mentioned studies published in the highest impact journals; however, within these publications, journalists opt for studies lower on the hierarchy of research design (i.e., observational studies rather than randomized controlled trials) [31,48]. With the coronavirus pandemic, journalists have been referencing more preprints (i.e., research that has not gone through the peer-review process) [49]. The influence of press releases on journalists and their coverage [48–50] is well established, as is journalists' desire to select studies that appeal to their audiences based on a mix of factors outside public or consumer health (notably, novelty and timeliness) [51].

Professional journalism practices and norms also influence the inclusion of links to research in news stories, an increasingly common practice [52,53]. For some journalists, links represent valuable additional sourcing [54] and credibility markers [53]; for others, they raise concerns over reader understanding, research literacy, and numeracy, as well as loss of readership and, in turn, advertising revenue to a publication [53]. Scholars attribute the pattern of traditional or legacy news sites directing readers to external links less often than digital-native news sites to these norms and practices, particularly the socialization of journalists within traditional professional journalism [55–62]. In a quantitative content analysis of stories from two top-tier prominent Belgian digital-native health news sites, researchers found that journalists linked to research from, on average, 30.9% of published articles, which is consistent with top US publications [62]. However, the frequency of these links varied by publication (33.9% versus 15.6%) [52,60,63–65]. In a content analysis of 270 blog posts and news-site articles (912 links), Coddington [60] found news sites had the highest percentage of posts (i.e., six or more links, 22%), though they were also most likely to have no links at all (36%), suggesting variance in linking practices among journalists [60]. Additionally, Coddington found that professional journalists do not typically link to external sites (9% of links versus 91% for internal links, i.e., those that point to a publication's own work), a finding aligned with previous research [62]. However, linking to research (classed as reference and fact-based information) was second only to linking to mainstream media content [60]. As Coddington explained, "news sites' links open the door to a valuable contextual resource for curious readers, but they reinforce a strict perimeter on the realm of acceptable discourse on public issues" [60].

Scientists work to influence journalists as a way to expand science communication more to the lay public [66]. They see journalists as a conduit for increasing the visibility of their work and recognize a professional responsibility to respond to journalists [66]. Scholars have suggested that issues between scientists and journalists— e.g., concerns over inaccurate reporting and misinformation, exaggeration, source selection, presentation of preprint research and bias — are gaining attention in an effort to improve the public's understanding of the scientific process [49,67]. Notably, Peters et al. [68] found that scientists and journalists interact more frequently and with less friction than previously reported. Yet, the researchers [66] also found that scientific communities have continued to regulate much of the news media contact with their members through norms that can be at odds with the influences and goals of public information departments. Scientists have a strategic view toward journalists and the marketing communications teams at both their institutions and the journals they publish in [66]. For instance, scientists have recognized the value journalists offer in terms of citation advantage

for studies mentioned in influential news publications [69,70]. They have also employed blogs and social media strategies for their research and scholarly practice (e.g., discourse, collaboration, recruitment), knowledge translation, dissemination, and education [71]. These strategies have often been visible to journalists and used to attract the attention of science journalists [72,73].

Findings from this study offer stakeholders— public health and communications professionals, as well as health professionals and journalists— a view into how online mentions of peer-reviewed journal articles funded by the US government appear in the news media. This study comes at a crucial time in light of current debates around science and a growing mistrust of the news media and concerns over fake news (i.e., rumors and falsehoods). While scientists are generally considered trusted as sources of information, their expertise is often met with public skepticism [74]. For instance, half or fewer Americans see science as inclusive of the best available evidence and they often doubt scientific consensus on issues considered to be generally agreed upon (i.e., effects of vaccines, causes of climate change, effects of eating genetically modified food) [74,75]. The National Science Foundation (NSF) Science and Engineering Indicators found that only about 31% of people in the US have a "clear understanding" of what constitutes a scientific study [75]. Journalists have historically been viewed as sense makers for helping the public understand scientific research. However, a report on public trust revealed that 42% of people in the US distrusted the media and 63% of people globally did not know how to tell quality journalism from fake news [76]. Additionally, 59% of people reported that it was becoming more difficult to tell if news had been produced by a reputable news organisation [76]. As such, our findings could facilitate future education (e.g., media and science literacy), dissemination and funding initiatives while describing the broad mix of news media now used by the public.

This study lays the groundwork for future research that explores how online news media could be better incorporated into dissemination processes and knowledge translation strategies. Understanding the professional practices and processes of how cancer research moves from scientists/researchers and marketing/communications professionals to journalists and then to the public is key for developing ways to more directly connect lay people with emerging scientific research in understandable and valuable ways.

## Spectrum of online news

Researchers' reliance on top-tier traditional news media with a national readership may be partly a result of practicality and commercial business models rather than the public's news and information habits. Databases of journalistic news content such as Lexisnexis and ProQuest can exclude popular digital-native news sites (e.g., Breitbart News Network, Buzzfeed, Mic), as well as popular niche sites (e.g., Business Insider, Bustle, Hello Giggles), which are typically considered outside mainstream journalism. These databases are also relatively static and do not capture the fluid nature of the news media landscape; they primarily act as archives of news stories and may not record all updates or changes to a story.

As an alternative to Lexisnexis and ProQuest, researchers have come to rely on Google News, an online aggregator of more than 4500 sources [44,77]. However, Google does not publicly release the sources included in Google News and not all content from news sites, notably those behind paywalls, are accessible through it [77,78]. Thus, transparent and replicable research using Google News can be challenging.

Altmetric LLC's database of more than 5000 global media sources is another alternative for accessing a broad mix of news sources. These sources are manually curated and are updated in real time through posts mentioning research via Application Programming Interfaces (APIs)

and Really Simple Syndication (RSS) feeds, software that allows the sharing of content through a standardized system for the distribution of content from online publishers. Altmetrics is particularly appropriate for scholarship considering the intersection of peer-reviewed research and journalists' portrayal of that research in online news [79]. Unlike Google News and Pro-Quest, Altmetric LLC was designed to identify, track and retrieve the contexts (e.g., news articles, Twitter, Facebook) in which research is mentioned [79]. Altmetric gathers news mentions of journal articles, including date and time of publication, from within media stories using unique identifying links to publications (i.e., hyperlinks or publication identifiers, such as Digital Object Identifiers, DOIs). Altmetric scans the text of media stories and uses natural language processing (NLP) techniques to amass study information (i.e., author names, journal titles, and study timeframes), which constitutes a "mention" [79].

Altmetric sources include well-known legacy news publications (e.g., *Los Angeles Times*, *New York Time*s, *Washington Post*), as well as digital-native publications (e.g., The Daily Beast, Huffington Post, Vox), popular niche sites (e.g., Dailyhunt, Politico, The Verge) and aggregation sites (e.g., EurekAlert, National Interest, Science Daily). Journalistic news media publications compose only a portion of Altmetric's source list, which also includes blogs, governmental documents, press release services and wire services. Altmetric is not without limitations, and like Google News, it does not publicly release a full list of sources; however, the company regularly shares its database and source list with researchers. Additionally, its list of media sources is not systematic, as source selection is based on manually curated news outlets, with content made available via third-party providers and RSS feeds.

This study characterises and analyses journalistic news stories that mention cancer research. It explores the intertwining hierarchy among an online collection of 86 traditional and digital-native news publications and their role in disseminating cancer research funded by the US government to the public. The study extends the work of Maggio et al. that identified patterns and frequencies of how 200264 scholarly studies about cancer circulated through mass media (e.g., press releases from news wires, non-profit and governmental media organisations, news aggregators and journalistic news organisations) [30]. Maggio et al. found that the frequency of journal articles about specific cancer types was misaligned with US rates of cancer burden (i.e., incidence and mortality) [30]. However, the researchers did not examine the alignment of journal articles specifically with journalistic news coverage, a popular [12–14,80] and more direct way the public accesses health information. This study addresses that gap.

## Materials and methods

This cross-sectional exploratory study builds on methods established by Maggio et al. [30], as a sub-analysis of the larger study, which examined scientific journal articles about cancer published in 2016 with US government funding sources. The present study 1) identifies, examines and characterizes how the number of mentions of research in online journalistic news sources varies by cancer type and publication; and 2) compares the disease incidence and mortality rates with the amount of research published for each cancer type and with the amount of news media attention each receives.

To facilitate transparency and replication of the methods, the project's complete data set and data management and analysis files are publicly accessible at https://zenodo.org/record/4075712 (see SPSS subfolder for project data and syntax files).

### Inclusion and exclusion criteria

We utilized Maggio et al.'s data set of journal articles published in 2016 to allow comparisons to findings regarding mass media coverage of cancer research [30]. By searching PubMed, the

researchers identified content about cancer (e.g., peer-reviewed papers, press releases, news stories, editorials, letters to the editor) via a query using Entrez Programming Utilities [30,81]. Maggio et al. included cancer articles as classified by National Library of Medicine (NLM) indexing. Within the search results, the researchers also identified articles regarding the 13 most frequently diagnosed cancer types in the United States (excluding non-melanoma skin cancers). The National Cancer Institute (NCI) identifies common cancer types as those with 40,000 or more cases reported annually [82]. These cancers, in order of estimated deaths, are lung, colon and rectum, pancreas, breast, liver, prostate, leukaemia, non-Hodgkin's lymphoma, bladder, kidney, endometrium, melanoma and thyroid.

Each journal article included citation metadata as provided by NLM (e.g., journal title, funding information, medical subject headings, or MeSH). Maggio et al. [30] categorized articles indexed with MeSH terms for a common cancer or respective child term as being about that particular cancer. For instance, articles indexed with "Leukemia, Myeloid" were classed with the parent term "Leukemia." Each cancer type mentioned in an article counted once toward its parent term. Articles without at least one of the 13 most prevalent cancers were classed as "other." Using the Altmetric database on 2 March 2018, Maggio et al. [30] searched each document's PubMed identifier and extracted those that received media attention through "mentions." Altmetrics characterized journal article "mentions" as the presence of either a link to a journal article or a phrase referencing a journal article in one of the 5000 sources included in its database.

## Codes and definitions

To generate a broad list of top US online journalistic media organizations, we combined two lists from Pew Research Center's 2016 State of the Media report. One list included the top 50 newspapers by average Sunday circulation and online presence [83]. The second list included the most popular digital-native sites, 36 in total [83]. We combined these lists and then used them to filter out non-journalistic news media sources from Maggio et al.'s [41] full data set, which contained 2805 media organizations, including non-journalistic media organizations (e.g., blogs, public relations and governmental agencies). This allowed for a data set with only online journalistic media sources. A journal article was coded as having a journalistic mention if it was cited in a news story published by at least one of the 86 journalistic news organizations included in the combined list from Pew. (For the names of the news media sources, see Pew Research Center [83]). Use of the combined list allowed us to generate results in a reproducible manner that can be re-examined for other years of publication. The combined lists composed 3.1% of the total number of media organizations contained in the Altmetric data set (86/2805). Other journalistic news outlets are part of the Altmetric data set; however, they are not part of this study. Additionally, we examined journalistic news coverage of scientific articles across annual estimated incidence and mortality totals for common types of cancer (i.e., defined by the National Cancer Institute as cancers with an estimated incidence of 40,000 or more cases per year) [84,85].

## Statistical analysis

Journal articles were coded for the presence or absence of at least one Almetric-defined mention in an online news story from the list of journalistic news outlets described above, as well as the total number of news mentions per article. We calculated descriptive statistics to provide a baseline understanding of the frequency and proportions of journal articles that received news mentions, as well as identify potential differences in the types of cancers covered by news

sources. Subsequently, we examined the characteristics of news organizations (e.g., news type, frequency) that provided coverage of these journal articles.

Two-tailed chi square analyses were conducted to compare coverage of the top 13 cancers across types of online news media sources (i.e., top traditional news sites versus top digital-native sites). An alpha level of .05 was the threshold for statistical significance. Analyses were conducted using Microsoft Excel 365 ProPlus (counts and figure creation) and SPSS version 21 (descriptives and chi-square tests).

## Results

As noted above, these analyses utilized a predefined data set from a previous study to facilitate direct comparisons with published research [30]. The data set included 11436 articles in PubMed published in 2016 that met predefined search and inclusion criteria, which included a cancer-related MeSH term (i.e., *neoplasm*) and at least one reported US funding source.

Within the original sample described above, 1.88% (n = 213) journal articles received at least one online news mention (i.e., instance of online news coverage). Of the 213 articles that received online news mentions, the median number of mentions per article was 1 (mean = 3; range: 1–23, SD = 3.8). Over half (n = 118, 55%) received one mention, 14% (n = 29) received two, and 31% (n = 66) had three or more mentions, for a total of 642 mentions across all 213 journal articles.

News mentions included journal articles from 96 scientific journals. News coverage most frequently cited research from *JAMA* (11 journal articles cited across 75 news mentions), *JAMA Internal Medicine* (7 articles, 50 mentions) and *New England Journal of Medicine* (8 articles, 49 mentions). The distribution of news mentions across journals varied in terms of both the total number of news mentions as well as the number of articles that received mentions. In other words, multiple news mentions could reflect a single journal article with extensive news coverage or multiple journal articles with lesser coverage per article. Thus, the *average* number of news mentions per journal article ranged from 1 (e.g., a journal with one journal article receiving one mention) up to 11 (*Journal of Psychopharmacology*), representing qualitatively distinct patterns of news coverage. Frequency of news mentions and cited journal articles among the top 10 scientific journals is presented in Table 1.

We conducted an exploratory analysis of a subset of news mentions (n = 236 news stories from all journal articles included in Table 4) to illustrate the content included in Altmetric's categorization of the story as a "mention." Briefly, these news stories included both stories

**Table 1. Top 10 research journals featuring the most US government-funded cancer research articles with news mentions in 2016.**

| Rank | Journal | Journal impact factor | Number of news mentions of cancer research articles in 2016 | Number of journal articles with 1 or more news mentions | Average news mentions per cancer research journal article |
|---|---|---|---|---|---|
| 1 | *JAMA* | 45.540 | 75 | 11 | 6.8 |
| 2 | *JAMA Int Med* | 18.652 | 50 | 7 | 7.1 |
| 3 | *N Engl J Med* | 74.699 | 49 | 8 | 6.1 |
| 4 | *Lancet* | 60.392 | 47 | 6 | 7.8 |
| 5 | *Science* | 41.845 | 34 | 5 | 6.8 |
| 6 | *Nature* | 42.778 | 33 | 6 | 5.5 |
| 7 | *Cancer* | 6.102 | 32 | 12 | 2.7 |
| 8 | *J Psychopharmacol* | 4.738 | 28 | 2 | 14 |
| 9 | *Proc Nat Acad Sci* | 9.412 | 26 | 14 | 1.9 |
| 10 | *BMJ* | 30.223 | 22 | 2 | 11 |

**Table 2. Common cancer types covered by journal articles in 2016 resulting from US government funds in relation to news and mass media mentions, number of estimated new cases in 2017 and estimated deaths.**

| Cancer type | No. of estimated new cases in 2017 | No. of estimated deaths in 2017 | No. of journal articles[a] n = 11436 (% of total sample) | No. of total online news mentions |
|---|---|---|---|---|
| Breast | 255190 | 41070 | 1284 (11.2) | 54 |
| Lung | 222500 | 155870 | 630 (5.5) | 35 |
| Melanoma | 87110 | 87110 | 302 (2.6) | 33 |
| Colon and rectal | 135430 | 50260 | 535 (4.7) | 28 |
| Prostate | 161360 | 26730 | 586 (5.1) | 23 |
| Leukemia | 62130 | 24500 | 544 (4.8) | 17 |
| Liver | 40710 | 28920 | 302 (2.6) | 8 |
| Pancreatic | 53670 | 43090 | 309 (2.7) | 5 |
| Endometrial | 61380 | 10920 | 77 (0.7) | 4 |
| Kidney | 63990 | 14400 | 106 (0.9) | 4 |
| Non-Hodgkin's lymphoma | 72240 | 20140 | 170 (1.9) | 3 |
| Thyroid | 56870 | 2010 | 71 (0.6) | 1 |
| Urinary/Bladder | 79030 | 16870 | 68 (0.6) | 0 |

*Articles may be counted multiple times if they include two or more cancers.

[a] Totals reported in this column are originally reported in Maggio et al., Table 7 [30].

focused on summaries of the journal article as well as mentions of the journal article as part of a news story on a broader topic. Of these news stories, 59.5% represented original content written by a journalist at the news outlet, while 40.5% of news stories originated from a wire service such as the Associated Press and United Press International (UPI). Eighty-two percent of news stories mentioned the name of the journal, such as *JAMA* or *BMJ*, and 74.9% mentioned the name of at least one co-author of the journal article. Most (69.0%) stories included a hyperlink to the specific journal article or its abstract on the journal's website.

All 13 of the most common cancers were included in the sample of journal articles. However, only 12 were the subject of at least one news mention (Table 2); articles addressing urinary and bladder cancer were not included in any identified news stories. Overall, 31% (n = 198) of the 642 news mentions included research addressing at least one of these common types of cancer. The majority of news mentions included research on only one cancer site, such as breast cancer or lung cancer (n = 187). Additionally, eight news mentions addressed two top cancers and three news mentions addressed three or more of these cancers. Frequency of news mentions differed across the common cancers, with breast cancer (n = 54), lung cancer (n = 35) and melanoma cancer (n = 33) being the subject of the most news mentions (Table 2). We also examined how mentions differed across estimated deaths for each common cancer type (Fig 1). These ratios ranged from 0 to 8618. With the exception of urinary and bladder cancers, which did not receive any mentions, the lowest death-to-mention ratio observed for breast cancer (ratio = 760.56, indicating greater coverage per estimated death), and the highest ratio observed for pancreatic cancer (ratio = 8618, indicating the least coverage per estimated death).

We also examined whether coverage of common cancer types differed by news source (traditional versus digital native). There was a significant association between news source type and mention of at least one common cancer type research (n = 642; X = 5.690, df = 1, $p$ = .017) with traditional news sources including more mentions of research on common cancer types (n = 127) compared to news mentions across digital native news sources (n = 71). Across 642

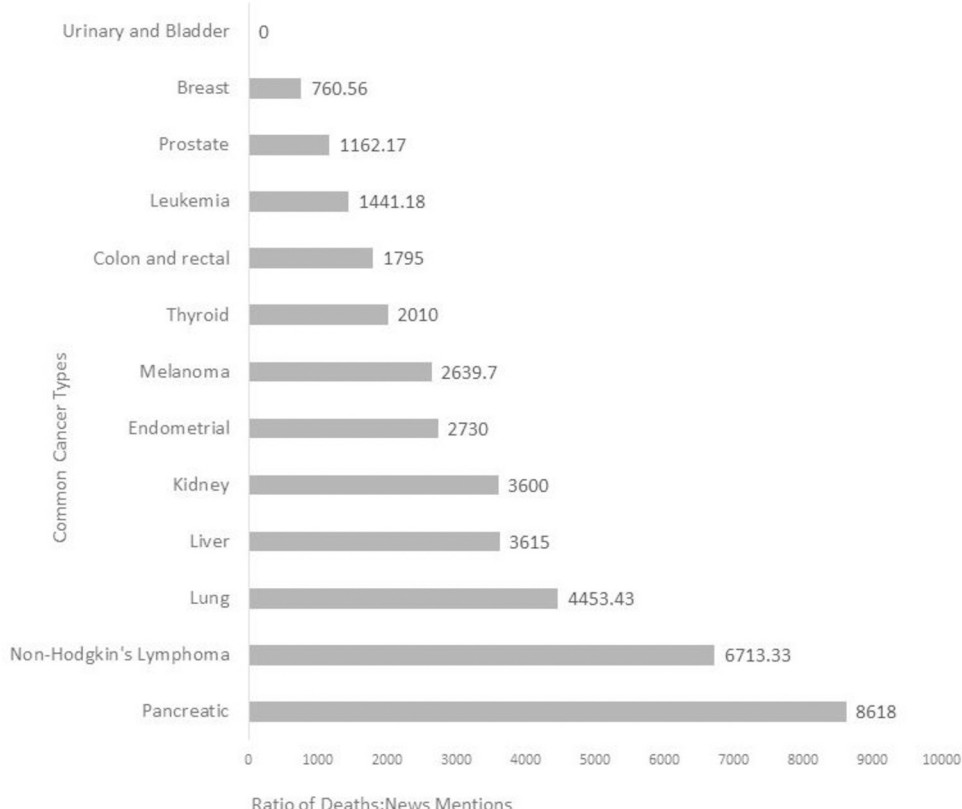

**Fig 1. Ratio of annual estimated deaths in 2017 to number of online news mentions in 2016 for common types of cancer.**

news mentions, 57.2% (n = 367) appeared in traditional online news sources, and 42.8% (n = 275) appeared in digital native news sources. In terms of the distribution of news coverage, the 642 online news mentions appeared across 53 online news sources, with an average number of 12 mentions per online news source (median = 5 mentions, range = 1–92). In total, 34 (68%) Pew-ranked newspapers mentioned at least one journal article from the sample, and 19 out of 36 sources (52.8%) from the Pew digital-native list mentioned at least one journal article. Five news sources accounted for approximately half of these mentions (see Table 4): *Philadelphia Inquirer* (n = 92), Business Insider (n = 70), *Washington Post* (n = 59), Huffington Post (n = 54), and *New York Times* (n = 50). The most frequently cited journal article, "Does physical activity attenuate, or even eliminate, the detrimental association of sitting time with mortality? A harmonised meta-analysis of data from more than 1 million men and women" (published in Lancet [30,86]), appeared in 23 online news mentions distributed across 16 news sources.

Yet, news sources (see Table 3) varied by audience size and reach. For instance, *Philadelphia Inquirer* (Philly.com) was the top news source by cancer research mentions, but had only 1.8 monthly unique visitors (MUVs), making it the lowest-ranked publication in terms of audience (*St. Louis Post-Dispatch* had 2 million MUVs and Breitbart News Network had 7.2 million MUVs). In contrast and in terms of cancer research was Huffington Post (68 million MUVs), Business Insider (75 million MUVs), *New York Times* (88 million MUVs) and *Washington Post* (90 million MUVs). The online news media landscape, based on the top 10 news sources by cancer research mentions, was divided equally between traditional and digital-native news.

**Table 3. Top 10 news sources by cancer research mentions in 2016.**

| Rank | News source | Type of news source | Monthly unique visitors, estimated* | Cancer research mentions in 2016 |
|---|---|---|---|---|
| 1 | *Philadelphia Inquirer* (Philly.com) | Traditional local news | 1.8 million | 92 |
| 2 | Business Insider | Digital-native news | 75 million | 70 |
| 3 | *Washington Post* | Traditional national news | 90 million | 59 |
| 4 | Huffington Post | Digital-native news | 68 million | 54 |
| 5 | *New York Times* | Traditional national news | 88 million | 50 |
| 6 | Breitbart News Network | Digital-native news | 7.2 million | 39 |
| 7 | International Business Times | Digital-native news | 21 million | 27 |
| 8 | *St. Louis Post-Dispatch* | Traditional local news | 2 million | 21 |
| 9 | *Houston Chronicle* | Traditional local news | 15 million | 20 |
| 10 | Quartz | Digital-native news | 22 million | 15 |

* Monthly unique visitors represent the number of distinct individuals requesting pages from a website during a given period; a unique visitor may visit a site multiple times within that same time frame, but will only count as one unique visitor [87–89].

This is notable, as there were more traditional than digital-native news sites in this sample (i.e., 50 versus 36, respectively). Of the five traditional news media publications, three were local publications. See Table 4 for a list of the top 20 journal articles by number of news mentions.

## Discussion

Journalistic news stories are a primary way the public accesses research [1–6]. In this study, only one to two (1.88) scientific journal articles out of every 100 mentioning US-funded cancer

**Table 4. Top journal articles by news mentions.**

| Rank | Journal | Journal impact factor | Title | Number of news mentions |
|---|---|---|---|---|
| 1 | *Lancet* | 60.392 | Does physical activity attenuate, or even eliminate, the detrimental association of sitting time with mortality? A harmonised meta-analysis of data from more than 1 million men and women. | 23*a |
| 2 | *Nature* | 42.778 | Substantial contribution of extrinsic risk factors to cancer development | 18* |
| 2 | *Science* | 41.845 | The phenotypic legacy of admixture between modern humans and Neanderthals | 18*a |
| 2 | *JAMA* | 45.54 | CDC guidelines for prescribing opioids for chronic pain | 18*a |
| 3 | *BMJ* | 30.223 | Overspending driven by oversized single dose vials of cancer drugs | 17 |
| 4 | *N Engl J Med* | 74.699 | Fulminant myocarditis with combination immune checkpoint blockade | 16* |
| 5 | *JAMA* | 45.54 | Comparison of Site of Death, Health Care Utilization, and Hospital Expenditures for Patients Dying With Cancer in 7 Developed Countries | 15* |
| 5 | *J Psychopharmacol* | 4.738 | Psilocybin produces substantial and sustained decreases in depression and anxiety in patients with life-threatening cancer: A randomized double-blind trial. | 15* |
| 6 | *JAMA Int Med* | 18.652 | Association of Specific Dietary Fats With Total and Cause-Specific Mortality. | 14* |
| 7 | *Cancer* | 5.742 | Effects of marital status and economic resources on survival after cancer: A population-based study. | 13*a |
| 7 | *J Psychopharmacol* | 4.738 | Rapid and sustained symptom reduction following psilocybin treatment for anxiety and depression in patients with life-threatening cancer: a randomized controlled trial. | 13*a |
| 7 | *N Engl J Med* | 74.699 | Regression of Glioblastoma after Chimeric Antigen Receptor T-Cell Therapy. | 13 |
| 8 | *JAMA* | 45.54 | US Spending on Personal Health Care and Public Health, 1996–2013. | 12* |
| 9 | *Science* | 41.845 | Mutational signatures associated with tobacco smoking in human cancer. | 11a |
| 10 | *J Clin Onc* | 32.956 | Financial Insolvency as a Risk Factor for Early Mortality Among Patients With Cancer. | 10 |
| 10 | *JAMA* | 45.54 | US County-Level Trends in Mortality Rates for Major Causes of Death, 1980–2014. | 10 |

* The journal issued a press release about the research.
a The journal article also ranked in the top 10 articles by mass media mentions as analyzed by Maggio, et al [30].

research received any online news attention. This is in contrast to the one out of every six journal articles (also reporting US-government funding) that received mass media attention, more broadly defined [30]. Our findings suggest that a great deal of cancer research goes unreported by both the online news media and the mass media, with journalists, in particular, acting as a gatekeeper for getting taxpayer-financed research to the general public. This is consistent with previous research finding that only a small proportion of available journal articles received coverage by the news media [90–92].

Findings revealed that health coverage of cancer research by journalists maps closely to the common cancer types covered by journal articles resulting from US government funds. The coverage highlights the ongoing issue of aligning news coverage and subsequent public access to research about cancers known to have the highest actual burden (i.e., incidence or mortality). This misalignment is notable, as prevention- and detection-focused research continues to be underrepresented in popular online news media.

A strength of this research is that it includes traditional (e.g., *Los Angeles Times*, *New York Times*, *Washington Post*) and local news sites (e.g., *Philadelphia Inquirer*, *St. Louis-Post Dispatch*, *Houston Chronicle*) alongside both digital-native news (e.g., Breitbart News, Buzzfeed, Vox) and niche news sites (e.g., *Business Insider*, *International Business Times*, Vice). When considered together, these sites offer a more complete view into the varied journalistic publications that report on cancer research in an evolving news media landscape. Findings highlight the importance of viewing news media penetration both holistically and by news media source.

Our study points to the continued dominance in cancer research coverage by traditional news organizations, but a dominance increasingly impinged by digital-native sites. While traditional news publications cited journal articles significantly more than digital-native and niche publications, the latter were not without mention among the top news sources. Five news sources accounted for approximately half of all journal article mentions (*Philadelphia Inquirer*, Business Insider, *Washington Post*, Huffington Post, *New York Times*), with two of these top sources being digital-native news sites (Business Insider and Huffington Post). Of the remaining top five, three were digital-native news.

The increasing popularity of digital-native news sites and their coverage of cancer studies suggest that researchers' tendency to focus on traditional news media may no longer suffice, particularly with the continued decline of traditional news organizations in terms of audience and financial stability [93]. Future research opportunities include exploring constraints that may affect the citation of research by news organizations. For instance, are digital-native and/or niche news organizations more restricted in their access of peer-reviewed research (e.g., less likely to gain early access to research from press departments of academic publishers that target major national news publications or less likely to pay costly fees to access academic journals) or are they simply catering to the content desires of their audiences? Additionally, do some of the professional practices at the different publications affect research coverage (e.g., one publication type hires more reporters with advanced science or health degrees)?.

The use of Altmetric allowed this study to include local publications, an often overlooked news source for health information. The top-ranked *Philadelphia Inquirer* stood out for its higher number of cancer research mentions (92 mentions in 2016, compared to the second-ranked Business Insider with 70). While multiple reasons likely contributed to *The Philadelphia Inquirer*'s high number of stories, a tendency to run licensed or wire stories (i.e., news copy sent out by news agencies to subscribers) may have influenced its rank. This tendency was shared by the Breitbart News Network, which appeared to run almost exclusively licensed stories mentioning cancer research. This is in contrast to the *Washington Post* and *New York Times* (ranked third and fifth, respectively), which typically ran articles based on original

reporting. These findings lay the groundwork for researchers to further explore the influence of news publications relying on licensed and news wire stories. Such an approach may suggest a consolidation of primary sources and a further culling of cancer research by the news media. Local publications are considered trustworthy and crucial providers of journalism in their communities [41]; however, the breakdown between news stories mentioning cancer research written by local reporters and those written by national and wire-service reporters is unclear. Documenting this breakdown would likely aid in reaching particular communities with cancer research and clarifying why local coverage may in fact be a local curation of national news and could suggest a lack of resources (e.g., a lack of access to journal articles or a shortage of health and science reporters at the local level). Additionally, public health and communication professionals and others interested in reaching local communities may benefit from our findings, which showed local publications to have a strong editorial interest in health research. A consideration about the size and characteristics of an audience could raise important questions for dissemination initiatives. For instance, is a mention in the *New York Times* the equivalent value of a mention in the *St. Louis Post-Dispatch*? While audience size suggests a difference in value, such a metric many not adequately take into account audience demographics, geography and community health needs.

As the results revealed, journalists and their news publications largely coalesced in terms of what cancer research they presented to the public, with 45% of journal articles receiving two or more mentions (for a total of 642 mentions across all 213 journal articles). Journalists also shared a tendency to report on research involving observational studies rather than RCTs, a finding aligned with previous studies [31,48]. This coming together in terms of what is newsworthy suggests that journalists, through professional practices (e.g., reporting, sourcing, etc.) and a shared value of top-tier brands, were largely united in what cancer research they considered relevant or of interest to the public.

Journalists, in particular, could benefit from greater context of their coverage of cancer research (e.g., how their research mentions map onto cancer incidence and mortality rates). Articles addressing urinary and bladder cancer were not included in any identified news stories, despite being sixth in terms of new cases among the 13 most common cancers included in US-funded journal articles. (Among these cancers, urinary and bladder ranks 10th in terms of estimated deaths.) However, breast cancer and melanoma were more heavily represented in news coverage relative to incidence and mortality rates, consistent with previous research [30]. Although melanoma is not among the top three cancers based on incidence or mortality, it ranked third based on news mentions—replacing prostate cancer. This contrasts with Maggio et al.'s [30] number of scientific articles about common cancer types in relation to media mentions. Breast, colon, lung, and prostate topped the list, followed by melanoma. The number of news mentions for melanoma was on par with that of lung cancer, despite lung cancer being responsible for most cancer deaths in the US each year. The reasons why melanoma may be mentioned in the news more frequently are unclear. However, melanoma may be discussed more because of seasonal coverage and sun-safety and anti-tanning campaigns. Also, former President Jimmy Carter announced in August 2015 that he had been diagnosed with metastatic melanoma, possibly leading journalists to pursue this cancer type more [94]. The influence of notable people and celebrities on the coverage of cancer by news publications is well reported in the literature [95]. Still, more research is needed into how journalists decide what types of cancer research are included in their reporting. For instance, when considering research for coverage, do journalists take into account cancer morbidity rates or do they rely more on other aspects (i.e., timeliness, celebrity news hook, journal venue or impact factor, press releases, scientists' institutional status or influence)?

Journalists most often mentioned research from journals with strong branding and marketing efforts, consistent with previous research [28]. For instance, the JAMA Network dominated the news mentions, with five of the top 20 articles. Of the 20 articles that ranked highest by news mentions, only five were not also associated with a press release though they may still have benefitted from promotional efforts by a journal. This aligns with Maggio et al.'s [30] finding that journals engaged in outreach and dissemination efforts received the most mass media coverage. It also raises the question, Are media professionals at risk of missing important research worthy of coverage because they hew to notoriety and the marketing efforts of certain journals? More research is needed into how marketing affects journalists' coverage of cancer research and if they are truly accessing the research (as opposed to working off an abstract or another media mention).

Journalists also appeared in agreement over mentioning journal articles that covered novel, sensational or surprising topics. In such cases, the inclusion of cancer research seemed to be of secondary importance. For instance, four of the top 20 journal articles by news mentions had impact factors below 6 (the median was 45.159). The research topics from these four studies (use of LSD, pasta consumption, and marital status effects on survival after cancer) were likely selected by journalists for their novelty rather than their public health importance. Additionally, two journals stand out for their high average of news mentions per journal article (a sign of intense news media attention): *Journal of Psychopharmacology* and *BMJ*. The former had two articles that garnered an average of 14 unique news mentions each, while the latter had two with an average of 11 unique news mentions each. The *Journal of Psychopharmacology* ran the following two articles: "Psilocybin produces substantial and sustained decreases in depression and anxiety in patients with life-threatening cancer: A randomized double-blind trial" [96] and "Rapid and sustained symptom reduction following psilocybin treatment for anxiety and depression in patients with life-threatening cancer: a randomized controlled trial" [97]. Journalists recast the titles of these articles into catchier headlines highlighting the novelty of the research. *New York Times* published two stories with the headlines: "A Dose of a Hallucinogen From a 'Magic Mushroom' and Then Lasting Peace" and "LSD to Cure Depression? Not So Fast." Huffington Post ran "Psychedelic Mushrooms And LSD Are Among The Safest Recreational Drugs, Survey Finds." The notable reach of these journal articles was likely due to their mention of a hallucinogenic alkaloid found in toadstools, rather than mention of cancer research. In the case of the top *BMJ* article, titled "Overspending driven by oversized single dose vials of cancer drugs" [98], journalists mentioned the article in 17 stories. *The Boston Globe* published "Study: $3B will be wasted on unused portion of cancer drugs" and *The Philadelphia Inquirer* ran "Overfilled cancer drug packs waste nearly $3 billion a year." The topic of public health finance and its appeal to taxpayers likely contributed to the journal article's reach, again, rather than mention of cancer research.

The top journal articles based on news mentions often aligned with the top journal articles based on mass media mentions [30]. For instance, the top journal articles for both news media and mass media mentions included five of the same articles, all of which were promoted with a press release. The overall top article, from the *Lancet* and about physical activity, garnered 462 mass media mentions but only 23 news mentions. This highlights the winnowing effect of journalists, but also the reach of a journal article that both taps into an almost universal health topic (e.g., physical activity) and offers an accessible title (i.e., Does physical activity attenuate, or even eliminate, the detrimental association of sitting time with mortality?). The other two articles, also with accessible titles— "CDC Guidelines for Prescribing Opioids for Chronic Pain" and "The phenotypic legacy of admixture between modern humans and Neanderthals"—highlight the value of tapping into a larger national conversation (i.e., the US opioid crisis and the Neanderthal ancestry reports promoted by companies such as 23andMe).

## Limitations

The findings of this study must be considered within the context of its limitations. Media mentions do not necessarily equate to actual readers. Nor does this study consider the accuracy and quality of news stories mentioning cancer research. Additionally, Altmetric's data set does not include all news publications; nor is it vetted externally. While PubMed actively indexes the US government funding for studies, authors are ultimately responsible for reporting funding. Also, the presented method includes only journal articles indexed in MEDLINE. Thus, we may have inadvertently missed including studies.

## Conclusions

Our findings map a formative landscape of the dissemination of federally funded cancer research across a spectrum of online news media organizations, including traditional and local news publications, as well as often overlooked digital-native news sources. This study shows that the coverage of cancer research by journalists differs from that generated by other, mass media news producers. Results revealed which news stories are most popular with the public online, as well as which journals and journal articles most typically appear in these stories. Findings highlight a continued misalignment between prevalent cancers and cancers highlighted in online news media, as well as a tendency by journalists to report on cancer research with a particularly surprising, entertaining or sensationalist bent.

This study has implications for funder groups (e.g., NCI, NIH) that might benefit from a replicable method for tracking characteristics from their research portfolio that appear in a broad mix of journalistic online news media. Findings can be used as a benchmark to evaluate future funding initiatives, dissemination efforts and knowledge transmission strategies. Public health communication and public relations professionals might also benefit from an examination of which characteristics of journal articles are most mentioned by journalists, as well as why cancer control and prevention receive less coverage from journalists than other cancer continuum stages.

A focus on journalistic news stories is crucial as it offers a more accurate view into health news penetration, as previous research included mass media mentions and may not accurately represent the general public's cancer news consumption or the changing newscape online. The role of journalist as gatekeeper is well established [46,99]. But less understood and still needing additional research is how journalists find, access, understand and use cancer research and how their news media organizations (e.g., traditional versus digital native, local versus national) and their professional practices (e.g., access to research through subscription or free databases, use of abstracts or full text, press briefings arranged by journals, direct contact with scientists) affect those behaviors [46].

As Brownson et al. [71] explained, ineffective dissemination contributes to a gap between the discovery of public health knowledge and its application in both policy development and the daily lives of people. Dissemination efforts may miss opportunities to consider what research reaches journalists and how they, in turn, decide what is worthy of coverage for a particular audience. While the media, collectively, are recognized as a key channel for knowledge dissemination [1–6], online digital-native news publications have received little attention, with researchers more focused on traditional legacy news publications. As such, a strength of this research is its board mix of online news organizations. As Maggio et al. reported [34], a broad inclusion of news sources is crucial as traditional news media are no longer the only or the primary sources of health information for the public. Future research should consider the intertwining relationship between scientists and journalists, particularly within social media and in

consideration of the promotion, coverage and framing of prevalent cancers alongside actual public cancer burden.

## Acknowledgments

The authors would like to thank Asura Enkhbayar and Juan Pablo Alperin for their methodological expertise associated with this paper.

## Disclaimer

The views expressed in this article are those of the authors and do not necessarily reflect the official policy or position of the Uniformed Services University of the Health Sciences, the Department of Defense, the National Cancer Institute, or the US Government.

## Author Contributions

**Conceptualization:** Laura Moorhead, Melinda Krakow, Lauren Maggio.

**Data curation:** Laura Moorhead, Melinda Krakow, Lauren Maggio.

**Formal analysis:** Laura Moorhead, Melinda Krakow, Lauren Maggio.

**Funding acquisition:** Lauren Maggio.

**Investigation:** Laura Moorhead, Melinda Krakow, Lauren Maggio.

**Methodology:** Laura Moorhead, Melinda Krakow, Lauren Maggio.

**Project administration:** Laura Moorhead, Melinda Krakow, Lauren Maggio.

**Writing – original draft:** Laura Moorhead, Melinda Krakow, Lauren Maggio.

**Writing – review & editing:** Laura Moorhead, Lauren Maggio.

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
