## [Decision Letter · Decision Letter 0]

27 Aug 2020

PONE-D-20-15200

What cancer research makes the news?

A quantitative analysis of online news stories that mention cancer studies

PLOS ONE

Dear Dr. Moorhead,

Thank you for submitting your manuscript to PLOS ONE. After careful consideration, we feel that it has merit but does not fully meet PLOS ONE’s publication criteria as it currently stands. Therefore, we invite you to submit a revised version of the manuscript that addresses the points raised during the review process.

We look forward to receiving your revised manuscript.

Kind regards,

Cindy Sing-bik Ngai

Academic Editor

PLOS ONE

Journal Requirements:

2. We noted in your submission details that a portion of your manuscript may have been presented or published elsewhere.

"See related manuscript, published in BMJ Open in 2018, uploaded with this submission. The data set from this previous manuscript was used for this paper; however, the effort does not constitute dual publication, as this paper explores a subset of data not highlighted in the previous study. "

Please clarify whether this or publication was peer-reviewed and formally published. If this work was previously peer-reviewed and published, in the cover letter please provide the reason that this work does not constitute dual publication and should be included in the current manuscript.

We note that one or more of the authors are employed by a commercial company: independent researcher.

3.1. Please provide an amended Funding Statement declaring this commercial affiliation, as well as a statement regarding the Role of Funders in your study. If the funding organization did not play a role in the study design, data collection and analysis, decision to publish, or preparation of the manuscript and only provided financial support in the form of authors' salaries and/or research materials, please review your statements relating to the author contributions, and ensure you have specifically and accurately indicated the role(s) that these authors had in your study. You can update author roles in the Author Contributions section of the online submission form.

3.2. Please also provide an updated Competing Interests Statement declaring this commercial affiliation along with any other relevant declarations relating to employment, consultancy, patents, products in development, or marketed products, etc.  

Reviewers' comments:

Reviewer's Responses to Questions

**Comments to the Author**

1. Is the manuscript technically sound, and do the data support the conclusions?

Reviewer #1: Partly

Reviewer #2: Partly

Reviewer #3: Yes

2. Has the statistical analysis been performed appropriately and rigorously? 

Reviewer #1: Yes

Reviewer #2: No

Reviewer #3: Yes

3. Have the authors made all data underlying the findings in their manuscript fully available?

Reviewer #1: Yes

Reviewer #2: Yes

Reviewer #3: Yes

4. Is the manuscript presented in an intelligible fashion and written in standard English?

Reviewer #1: Yes

Reviewer #2: Yes

Reviewer #3: Yes

5. Review Comments to the Author

Reviewer #1: This paper can be considered as a subsection of a previously published paper in which an overview of altmetric mentions to US funded papers on cancer research. In this case, the authors focus on news stories. The paper is pretty straightforward and has no technical complications. While the motivation of the paper is of interest, the authors do not feed much from scientific communication literature on interests of scientists communicating with journalists, motivations, etc. There is a vast stream of literature on this which would greatly enrich, both the introduction as well as the discussion.

The analysis the authors make is quite superficial without deepening on motivations, external factors that may affect being mentioned in news media (e.g., journal venue, press releases, authors' institutional status, authors' influence), and I would say this is more of an exploratory paper than anything else.

So in general I find it quite poor as it does not delve much into the richness of the data they have nor they go beyond a basic descriptive analysis.

Beyond that, there are two specific sentences the authors make I do not agree and should be modified if accepted for publication later on.

- They indicate that partly, their novelty is on the journalists' interest on funded cancer research and use Altmetric.com as a 'better source' than others because it includes online news media. While this may be partly true, the authors ignore in the text two important limitations of this source: 1) the list of news media is quite arbitrary. The link the authors provided no longer refers to the list of news media from Altmetric.com. this should be updated. 2) News media mentions are identified by hyperlinks to papers, which is something that not always happens when reporting research in news media. This may affect especially traditional media which has a lower online presence and may not include hyperlinks to scientific papers, hence the differences in the results.

- In page 4, paragraph 3 the authors state the following: 'Increasingly, researchers utilize Altmetric’s database of more than 2500 global media sources'. There is no evidence of this whatsoever and no references are given.

Reviewer #2: Review of PONE-D-20-15200

This is a mostly descriptive paper about coverage of cancer topics in the media; the topic of this paper is important and timely. The introduction and discussion are interesting. However, the methods and results were underdeveloped. This might be addressed by clarifying some of the definitions and how variables were measured/coded. Perhaps providing a few examples of what was coded as a mention, or adding a list of mentions for a couple of the top mentioned papers would help. I would recommend removing the chi-squared analyses and, instead, creating some visuals that demonstrate the relative differences for incidence/death/mentions by cancer type and media type. Attaching one example of how this might look. I think a set of graphs would be a lot more powerful than the lists of numbers currently included.

Some information about the size and characteristics of the audience of the different outlets could add to the understanding of the reach of the different types of cancer information. Is a mention in the NYT equivalent in audience reach to a mention in the St. Louis Post-Dispatch, for example. These numbers may be tricky to get, but there are likely estimates of audience or market share.

In addition, given the current political climate and description of many of these outlets as fake news and growing public distrust of science, it seems like the inclusion of science in a broad spectrum of media outlets is extremely important. Some discussion of these topics could be useful.

Finally, there was some discussion of how journalists find science to report on and, from the lists shown in the paper, it seems that journal impact factor/visibility is probably a big part of it. Academics and academic institutions have been more visible and active on social media in recent years, which could influence the reporting of science if academics/academic institutions share science this way and “tag” journalists or journalistic outlets. (https://www.sciencedirect.com/science/article/pii/S2211419X2030029X)

Other possible edits:

- First two sentences of last paragraph in the abstract are confusing, reword to clarify.

- The files included are the data and data collection files, but the data management and analysis files are not available at the currently provided link. Including the data is great, but the paper is not reproducible without the statistical code as well. Use of Microsoft Excel can be problematic for reproducibility (see https://www.washingtonpost.com/news/wonk/wp/2016/08/26/an-alarming-number-of-scientific-papers-contain-excel-errors/ and Ziemann M, Eren Y, El-Osta A. Gene name errors are widespread in the scientific literature. Genome biology. 2016 Dec;17(1):1-3.

- Chi-squared can only find associations, not the direction of association, so this sentence needs to be re-worded or an analysis of the standardized residuals should be included to support the finding: “Traditional news sources included significantly more mentions of research on common cancer types (n = 240) compared to news mentions across digital native news sources (n = 204; X = 5.690, df = 1, p = .017).” One suggestion for rewording would be, “There was a significant association between news source type and mention of common cancer type research (n = 204; X = 5.690, df = 1, p = .017) with traditional news sources including more mentions of research on common cancer types (n = 240) compared to news mentions across digital native news sources.” It is a subtle distinction, but important given how chi-squared is computed. Following up with standardized residuals to determine which of the frequencies in the chi-squared were much different from expected would strengthen the results section and perhaps provide the authors and readers with additional insights.

- Including the IQR in addition to the range would be helpful in understanding the data. Or, as suggested above, including the statistical code so that interested readers could examine the distribution of the mentions per online news source.

- The standard deviation being higher than the mean, along with the range being so wide for number of mentions, suggests that this distribution is skewed and the median should be reported instead. It looks like the median is 1, so the mean of 3 is definitely exaggerating the central tendency.

- In table 1 it might be useful to add some sort of mentions/death or death/mention metric; it takes some work as the table currently is formatted to understand that, for example, pancreatic cancer is woefully under-reported given the amount of death (more than breast cancer! I had no idea.) Or, alternatively, a visual that compares the mortality rank and publicity rank or something similar so that this disconnect between incidence/mortality and publicity are more clear. …as a journalist might say, it seems like the authors have buried the lead.

- The column headings on Table 2 are really confusing; please clarify. Also, add a date range for the articles to the title of this table or to the “Total news mentions” column heading.

***Table1data to make graph (put in a csv to use R code below)***

cancer incidence deaths num.articles mentions

Breast 255190 41070 1284 54

Lung 222500 155870 630 35

Melanoma 87110 87110 302 33

Colon and rectal 135430 50260 535 28

Prostate 161360 26730 586 23

Leukemia 62130 24500 544 17

Liver 40710 28920 302 8

Pancreatic 53670 43090 309 5

Endometrial 61380 10920 77 4

Kidney 63990 14400 106 4

Non-Hodgkin’s 72240 20140 170 3

Thyroid 56870 2010 71 1

Urinary/Bladder 79030 16870 68 0

***R code for graph***

# open data

table1 <- read.csv("table1.csv")

# load tidyverse for graphing

library(package = "tidyverse")

# make graph of mentions & deaths

longer <- table1 %>%

mutate(deathsInThousands = deaths/1000) %>%

pivot_longer(cols = c("deathsInThousands","mentions"),

names_to = c("metric"),

values_to = "freqNum") %>%

drop_na() %>%

mutate(metric = as.factor(metric))

longer %>%

ggplot(aes(x = reorder(cancer, freqNum), y = freqNum, fill = metric)) +

geom_col(position = "dodge") +

coord_flip() +

theme_minimal() +

labs(y = "Frequency", x = "Cancer type")

Reviewer #3: Interesting article and approach! Here are some questions or suggestions overall:

Introduction:

1. It wasn't completely clear what "journalistic media" meant. A more concrete definition within the Introduction would be helpful as you dig down into the Methods, perhaps in lieu of the one you provided (e.g., includes both print and online sources). For example, in the paragraph beginning with: "As Maggio et al.’s [41] full data set included a broad collection of news media organizations, we filtered out non-journalistic news media sources from the data set, leaving only online news media sources." What is a non-journalistic media source? What exactly was filtered out? A clearer definition (perhaps with examples) would be helpful.

Methods:

1. In general, a little more detail or clarity about your process with Altmetric would be helpful, particularly for those who have never used the platform before. For example, you write: "The combined lists composed 3.1% of the total Altmetric data set (86/2805)." I'm not sure what the 2805 is referring to or how that number was obtained.

2. Similarly, you describe coding articles for the presence of a mention. As someone who is unfamiliar with Altmetric, was coding an automated process, or was this done manually? If the latter, more information about how this was done would be helpful.

3. Extensive detail provided regarding how the media-related data were obtained. However, a brief mention of where incidence/mortality data were derived from would be helpful, too, since that's a major aim of the paper.

4. It's nice to have all coding for this project publicly available!

Results:

1. Minor issue but Table 2 is referenced in text before Table 1.

2. Table 1 was particularly interesting, but little reporting or discussion of it was presented in the text. If part of the goal of this paper is to highlight discrepancies between morbidity/mortality and news coverage, I might highlight some "standouts" in the text. For example, lung cancer is responsible for ~150,000 deaths annually and received 35 online mentions, while melanoma (responsible for half as many deaths) received nearly identical coverage. A greater discussion of these points would serve to support your overall study aim.

Discussion:

1. The overall organization of the Discussion section made it a bit difficult to follow at times. In its current form, it seems to jump around, and it’s difficult to see how the findings of the present study fit with other relevant research. The structure proposed in the BMJ (Docherty & Smith, 1999) may be useful, and I encourage the authors to consider using it in this paper. doi: https://doi.org/10.1136/bmj.318.7193.1224

2. There are portions of the Discussion section that would be better placed within the Results. (Example: First paragraph under the "Growing divide: traditional vs. digital-native news" heading)

3. You mention that certain journal article topics (e.g., drugs or financing) are prioritized, it would be interesting to see what situated within the context of previous research as well. I would assume that this is common practice, but is that something unique to this study, or is that aligned with previous research on the topic?

4. Something that's missing from this section is a discussion of how this work relates to intentional dissemination efforts from researchers. Within the Introduction, you write: "Findings could facilitate future dissemination and funding initiatives... This study lays the groundwork for future research that explores how online news media could be better incorporated into dissemination processes and knowledge translation strategies." You also cite the 2018 Brownson article but don't discuss it or any related articles within this section. More consideration within the Discussion is warranted, as it seems to be a logical "next step" for this type of work.

6. PLOS authors have the option to publish the peer review history of their article (what does this mean?). If published, this will include your full peer review and any attached files.

Reviewer #1: No

Reviewer #2: **Yes: **Jenine Harris

Reviewer #3: No

---

## [Author Response · Author response to Decision Letter 0]

15 Oct 2020

[[Please see the uploaded letter for a better formatted version of the following.]]

Dear editors,

We thank you and the reviewers for your feedback and suggestions regarding our manuscript entitled, "What cancer research makes the news? A quantitative analysis of online news stories that mention cancer studies" (Manuscript PONE-D-20-15200) by Moorhead, Krakow and Maggio. We are pleased to resubmit our manuscript with revisions based on this feedback. In the table below, we provide a description of how we addressed each reviewer comment. These revisions are also noted within the document using track changes.

Editor comments

Author response

Edit location

Thank you. We have verified that we have done this.

NA

2. We noted in your submission details that a portion of your manuscript may have been presented or published elsewhere.

"See related manuscript, published in BMJ Open in 2018, uploaded with this submission. The data set from this previous manuscript was used for this paper; however, the effort does not constitute dual publication, as this paper explores a subset of data not highlighted in the previous study. "

Please clarify whether this or publication was peer-reviewed and formally published. If this work was previously peer-reviewed and published, in the cover letter please provide the reason that this work does not constitute dual publication and should be included in the current manuscript.

Again, this paper explores a subset of data not examined in the previous study; as such it does not constitute dual publication. The previous paper was peer-reviewed and published in BMJ Open. The citation is as follows: 

Maggio, L. A., Ratcliff, C. L., Krakow, M., Moorhead, L. L., Enkhbayar, A., & Alperin, J. P. (2019). Making headlines: an analysis of US government-funded cancer research mentioned in online media. BMJ open, 9(2), e025783.

NA

3.1. Please provide an amended Funding Statement declaring this commercial affiliation, as well as a statement regarding the Role of Funders in your study. If the funding organization did not play a role in the study design, data collection and analysis, decision to publish, or preparation of the manuscript and only provided financial support in the form of authors' salaries and/or research materials, please review your statements relating to the author contributions, and ensure you have specifically and accurately indicated the role(s) that these authors had in your study. You can update author roles in the Author Contributions section of the online submission form. 

3.2. Please also provide an updated Competing Interests Statement declaring this commercial affiliation along with any other relevant declarations relating to employment, consultancy, patents, products in development, or marketed products, etc. 

We have no funding and/or competing interests to declare. Apologies for any confusion on this front.

NA

Reviewers’ Responses to Questions

1. Is the manuscript technically sound, and do the data support the conclusions?

Reviewer #1: Partly

Reviewer #2: Partly

Reviewer #3: Yes

Throughout the manuscript, we have worked to improve the support of our conclusions. These changes include clarifying variable definitions/codes and methodological description and expanding our statistical analyses to address reviewer comments. 

Throughout manuscript

2. Has the statistical analysis been performed appropriately and rigorously?

Reviewer #1: Yes

Reviewer #2: No

Reviewer #3: Yes

We have updated our statistical analyses to address comments raised by Reviewer 2. 

Results section, pp. 4–11.

3. Have the authors made all data underlying the findings in their manuscript fully available?

Reviewer #1: Yes

Reviewer #2: Yes

Reviewer #3: Yes

To facilitate transparency and replication of the methods, the project’s complete data set and data management and analysis files are publicly accessible at https://doi.org/10.5281/ zenodo.1306984.

p. 5

4. Is the manuscript presented in an intelligible fashion and written in standard English?

Reviewer #1: Yes

Reviewer #2: Yes

Reviewer #3: Yes

NA

NA

Reviewer Comments to the Author

Reviewer #1

This paper can be considered as a subsection of a previously published paper in which an overview of altmetric mentions to US funded papers on cancer research. In this case, the authors focus on news stories. The paper is pretty straightforward and has no technical complications. While the motivation of the paper is of interest, the authors do not feed much from scientific communication literature on interests of scientists communicating with journalists, motivations, etc. There is a vast stream of literature on this which would greatly enrich, both the introduction as well as the discussion.

The analysis the authors make is quite superficial without deepening on motivations, external factors that may affect being mentioned in news media (e.g., journal venue, press releases, authors' institutional status, authors' influence), and I would say this is more of an exploratory paper than anything else. 

Thank you for the suggestion. We have incorporated additional information from scientific communication literature regarding scientists communicating with journalists.

We agree that this paper is primarily exploratory, as there is a gap in existing literature about Altmetric mentions specific to US-funded papers on cancer research by journalists. However, with this revision, we have worked to provide greater description of the methods and analyses in order to provide a solid foundation for future research building off of these exploratory findings.

pp. 2–3, 14 

Beyond that, there are two specific sentences the authors make I do not agree and should be modified if accepted for publication later on.

- They indicate that partly, their novelty is on the journalists' interest on funded cancer research and use Altmetric.com as a 'better source' than others because it includes online news media. While this may be partly true, the authors ignore in the text two important limitations of this source: 1) the list of news media is quite arbitrary. The link the authors provided no longer refers to the list of news media from Altmetric.com. this should be updated. 2) News media mentions are identified by hyperlinks to papers, which is something that not always happens when reporting research in news media. This may affect especially traditional media which has a lower online presence and may not include hyperlinks to scientific papers, hence the differences in the results.

We have addressed the issues with these two specific sentences in our revision. We have revised and expanded out the section on Altmetric.com, including how Almetric uses both hyperlinks and phrases to scientific paper. 

pp. 4–5

- In page 4, paragraph 3 the authors state the following: 'Increasingly, researchers utilize Altmetric’s database of more than 2500 global media sources'. There is no evidence of this whatsoever and no references are given.

This issue has been addressed through the revision of this paragraph. Additional citations have been added, along with six lines regarding the limitations of using Altmetric’s database.

pp. 4–5

Reviewer #2

This is a mostly descriptive paper about coverage of cancer topics in the media; the topic of this paper is important and timely. The introduction and discussion are interesting. However, the methods and results were underdeveloped. This might be addressed by clarifying some of the definitions and how variables were measured/coded. Perhaps providing a few examples of what was coded as a mention, or adding a list of mentions for a couple of the top mentioned papers would help. I would recommend removing the chi-squared analyses and, instead, creating some visuals that demonstrate the relative differences for incidence/death/mentions by cancer type and media type. Attaching one example of how this might look. I think a set of graphs would be a lot more powerful than the lists of numbers currently included.

Thank you for the suggestion to help clarify our definitions and coding methods for the reader as well as illustrate our analyses more clearly. To address this comment, we have first expanded the description of our analytic variables in the methods section, including more description of Altmetric’s definition of a media mention, as well as a definition and reference for common types of cancers. 

We have also added Figure 1 to provide a visualization of relative differences in the ratio of estimated deaths per news mention across all common types of cancer. We provide a brief description to accompany this figure on pages 6-7 as well, which now reads: 

“We also examined how mentions differed across estimated deaths for each common cancer type (Figure 1). These ratios ranged from 0 to 8618. With the exception of urinary and bladder cancers, which did not receive any mentions, the lowest death to mention ratio observed for breast cancer (ratio = 760.56, indicating greater coverage per estimated death), and the highest ratio observed for pancreatic cancer (ratio = 8618, indicating the least coverage per estimated death).”

pp. 6–7, 4–12

Some information about the size and characteristics of the audience of the different outlets could add to the understanding of the reach of the different types of cancer information. Is a mention in the NYT equivalent in audience reach to a mention in the St. Louis Post-Dispatch, for example. These numbers may be tricky to get, but there are likely estimates of audience or market share.

Thank you for this suggestion. We have added a column to Table 3 (p. 9) that incorporates the estimated monthly unique visitors for each publication, as an indicator of audience size and characteristics. Additionally, the Discussion (pp. 12–15) now includes reference to the differences between local and national news, as well as the changing digital media landscape. 

pp. 9, 12–15

In addition, given the current political climate and description of many of these outlets as fake news and growing public distrust of science, it seems like the inclusion of science in a broad spectrum of media outlets is extremely important. Some discussion of these topics could be useful.

We appreciate this suggestion and have now incorporated this into both the Introduction (p. 4) and the Discussion (pp. 12–15). 

pp. 12–15

Finally, there was some discussion of how journalists find science to report on and, from the lists shown in the paper, it seems that journal impact factor/visibility is probably a big part of it. Academics and academic institutions have been more visible and active on social media in recent years, which could influence the reporting of science if academics/academic institutions share science this way and “tag” journalists or journalistic outlets. (https://www.sciencedirect.com/science/article/pii/S2211419X2030029X)

Thank you for this point. Table 2 (p. 9) includes the impact factor for each journal and it is considered in the Discussion (pp. 14). 

pp. 9, 14

Other possible edits:

- First two sentences of last paragraph in the abstract are confusing, reword to clarify.

We have rewritten these two sentences to address the reviewer’s concern.

p. 1

- The files included are the data and data collection files, but the data management and analysis files are not available at the currently provided link. Including the data is great, but the paper is not reproducible without the statistical code as well. Use of Microsoft Excel can be problematic for reproducibility (see https://www.washingtonpost.com/news/wonk/wp/2016/08/26/an-alarming-number-of-scientific-papers-contain-excel-errors/ and Ziemann M, Eren Y, El-Osta A. Gene name errors are widespread in the scientific literature. Genome biology. 2016 Dec;17(1):1-3.

We have addressed this: https://zenodo.org/record/4075712

NA

- Chi-squared can only find associations, not the direction of association, so this sentence needs to be re-worded or an analysis of the standardized residuals should be included to support the finding: “Traditional news sources included significantly more mentions of research on common cancer types (n = 240) compared to news mentions across digital native news sources (n = 204; X = 5.690, df = 1, p = .017).” One suggestion for rewording would be, “There was a significant association between news source type and mention of common cancer type research (n = 204; X = 5.690, df = 1, p = .017) with traditional news sources including more mentions of research on common cancer types (n = 240) compared to news mentions across digital native news sources.” It is a subtle distinction, but important given how chi-squared is computed. Following up with standardized residuals to determine which of the frequencies in the chi-squared were much different from expected would strengthen the results section and perhaps provide the authors and readers with additional insights.

Thank you to the reviewer for noting this important distinction regarding chi-squared tests and providing a suggested revision to offer a clearer interpretation of the results, given the limitations of this statistical test. We have adopted the revised wording suggested by the reviewer, and the lines on page 7 now read: 

“There was a significant association between news source type and mention of common cancer type research (n = 204; X = 5.690, df = 1, p = .017) with traditional news sources including more mentions of research on common cancer types (n = 240) compared to news mentions across digital native news sources.”

p. 7

- Including the IQR in addition to the range would be helpful in understanding the data. Or, as suggested above, including the statistical code so that interested readers could examine the distribution of the mentions per online news source.

Following the suggestion of the reviewers, we have included a copy of our statistical code and dataset for interested readers to examine. 

NA

- The standard deviation being higher than the mean, along with the range being so wide for number of mentions, suggests that this distribution is skewed and the median should be reported instead. It looks like the median is 1, so the mean of 3 is definitely exaggerating the central tendency.

Thank you for this suggestion. To more accurately represent the distribution of mentions, we have updated this section to report the median, with the mean and range included in parentheses. This sentence on page 6 now reads: 

“Of the 213 articles that received online news mentions, the median number of mentions per article was 1 (mean = 3; range: 1-23, SD=3.8).”

p. 6

- In table 1 it might be useful to add some sort of mentions/death or death/mention metric; it takes some work as the table currently is formatted to understand that, for example, pancreatic cancer is woefully under-reported given the amount of death (more than breast cancer! I had no idea.) Or, alternatively, a visual that compares the mortality rank and publicity rank or something similar so that this disconnect between incidence/mortality and publicity are more clear. …as a journalist might say, it seems like the authors have buried the lead.

We appreciate the reviewer’s suggestion to include a visual to illustrate the ratio of estimated annual deaths to news mentions. The suggestion of this metric is indeed helpful in demonstrating the variation in coverage across types of cancers, given their mortality estimates. To address this, we have added the following text on pages 6–7: 

“We also examined how mentions differed across estimated deaths for each common cancer type (Figure 1). These ratios ranged from 0 to 8618. With the exception of urinary and bladder cancers, which did not receive any mentions, the lowest death to mention ratio observed for breast cancer (ratio = 760.56, indicating greater coverage per estimated death), and the highest ratio observed for pancreatic cancer (ratio = 8618, indicating the least coverage per estimated death). “

We have also added Figure 1 to provide a visualization of these ratios as well. 

pp. 6–7

- The column headings on Table 2 are really confusing; please clarify. Also, add a date range for the articles to the title of this table or to the “Total news mentions” column heading.

Thank you for this suggestion. We have revised the column headings for columns 3-5 greater clarity and also added the year to the table’s title. 

pp. 7–12

***Table1data to make graph (put in a csv to use R code below)***

cancer incidence deaths num.articles mentions

Breast 255190 41070 1284 54

Lung 222500 155870 630 35

Melanoma 87110 87110 302 33

Colon and rectal 135430 50260 535 28

Prostate 161360 26730 586 23

Leukemia 62130 24500 544 17

Liver 40710 28920 302 8

Pancreatic 53670 43090 309 5

Endometrial 61380 10920 77 4

Kidney 63990 14400 106 4

Non-Hodgkin’s 72240 20140 170 3

Thyroid 56870 2010 71 1

Urinary/Bladder 79030 16870 68 0

***R code for graph***

# open data

table1 <- read.csv("table1.csv")

# load tidyverse for graphing

library(package = "tidyverse")

# make graph of mentions & deaths

longer <- table1 %>%

mutate(deathsInThousands = deaths/1000) %>%

pivot_longer(cols = c("deathsInThousands","mentions"),

names_to = c("metric"),

values_to = "freqNum") %>%

drop_na() %>%

mutate(metric = as.factor(metric))

longer %>%

ggplot(aes(x = reorder(cancer, freqNum), y = freqNum, fill = metric)) +

geom_col(position = "dodge") +

coord_flip() +

theme_minimal() +

labs(y = "Frequency", x = "Cancer type")

Thank you for sharing this code with us. Although the present analysis was not conducted in R, this is quite helpful to view for our future work. In the spirit of the reviewer’s comment, we have used Excel to generate Figure 1 to illustrate a comparison of estimated deaths per news mentions across common cancer types. 

NA

Reviewer #3

Interesting article and approach! 

Thank you for your comment.

1. It wasn't completely clear what "journalistic media" meant. A more concrete definition within the Introduction would be helpful as you dig down into the Methods, perhaps in lieu of the one you provided (e.g., includes both print and online sources). For example, in the paragraph beginning with: "As Maggio et al.’s [41] full data set included a broad collection of news media organizations, we filtered out non-journalistic news media sources from the data set, leaving only online news media sources." What is a non-journalistic media source? What exactly was filtered out? A clearer definition (perhaps with examples) would be helpful.

We added a more concrete definition in the Introduction (p. 3) and clarified in the Methods section that we relied on Pew Research Center’s 2016 State of the Media Report for our list of journalistic media (p. 6).

pp. 3, 6

Methods:

1. In general, a little more detail or clarity about your process with Altmetric would be helpful, particularly for those who have never used the platform before. For example, you write: "The combined lists composed 3.1% of the total Altmetric data set (86/2805)." I'm not sure what the 2805 is referring to or how that number was obtained.

Thank you for suggesting this clarification. We have expanded this description to clarify that the 2805 refers to the total number of all media organizations included in the Altmetric data, which we then narrowed down to a focused list of 86 journalistic news organizations for the present analyses. This section on page 6 now reads: 

“We combined these lists and then used them to filter out non-journalistic news media sources from Maggio et al.’s [41] full data set, which included 2805 media organizations, including non-journalistic media organizations (e.g., blogs, public relations and governmental agencies). This allowed for a data set with only online journalistics media sources. A journal article was coded as having a journalistic mention if it was cited in a news story published by at least one of the 86 journalistic news organizations included in from these) two combined lists from Pew. (For the names of the news media sources, see Pew Research Center [45]). Use of these combined lists allowed us to generate results in a reproducible manner that can be re-examined for other years of publication. The combined lists composed 3.1% of the total number of media organizations contained in the Altmetric data set (86/2805).”

p. 6

2. Similarly, you describe coding articles for the presence of a mention. As someone who is unfamiliar with Altmetric, was coding an automated process, or was this done manually? If the latter, more information about how this was done would be helpful.

We have addressed this concern in our revision. We have revised and expanded out the section on Altmetric.com, including how Almetric codes its news sources. 

pp. 4–5

3. Extensive detail provided regarding how the media-related data were obtained. However, a brief mention of where incidence/mortality data were derived from would be helpful, too, since that's a major aim of the paper.

Thank you for this suggestion. On page 6, we have added wording to mention where data on cancer incidence and mortality were obtained and added a citation to the Common Types of Cancer list provided by the National Cancer Institute. This section now reads: 

“Additionally, we examined journalistic news coverage of scientific articles across common types of cancer (i.e., defined by the National Cancer Institute as cancers with an estimated incidence of 40,000 or more cases per year), which included breast, lung, melanoma, colorectal, prostate, leukemia, liver, pancreatic, endometrial, kidney, Non-Hodgkin’s lymphoma, thyroid, and urinary/bladder cancers."

p. 6

4. It's nice to have all coding for this project publicly available!

Thank you. We agree. 

Results:

1. Minor issue but Table 2 is referenced in text before Table 1.

We have addressed this.

pp. 7–9

2. Table 1 was particularly interesting, but little reporting or discussion of it was presented in the text. If part of the goal of this paper is to highlight discrepancies between morbidity/mortality and news coverage, I might highlight some "standouts" in the text. For example, lung cancer is responsible for ~150,000 deaths annually and received 35 online mentions, while melanoma (responsible for half as many deaths) received nearly identical coverage. A greater discussion of these points would serve to support your overall study aim.

Thank you for this suggestion. We have expanded our discussion of Table 1 in the Discussion and Conclusion. We have also worked it better into the Abstract.

pp. 1, 4–15

Discussion:

1. The overall organization of the Discussion section made it a bit difficult to follow at times. In its current form, it seems to jump around, and it’s difficult to see how the findings of the present study fit with other relevant research. The structure proposed in the BMJ (Docherty & Smith, 1999) may be useful, and I encourage the authors to consider using it in this paper. doi: https://doi.org/10.1136/bmj.318.7193.1224

Thank you for pointing us to this paper. You will see that we incorporated the suggested structure into the discussion. In the Track Changes version, you can see the suggested structure via subheads, which was removed in the “clean” version.

pp. 12–16

2. There are portions of the Discussion section that would be better placed within the Results. (Example: First paragraph under the "Growing divide: traditional vs. digital-native news" heading)

We moved that initial paragraph and several other portions of the Discussion section into the Results section. 

p. 6

3. You mention that certain journal article topics (e.g., drugs or financing) are prioritized, it would be interesting to see what situated within the context of previous research as well. I would assume that this is common practice, but is that something unique to this study, or is that aligned with previous research on the topic?

We have tried to address this in the literature section (i.e., Introduction). 

pp. 1–6

4. Something that's missing from this section is a discussion of how this work relates to intentional dissemination efforts from researchers. Within the Introduction, you write: "Findings could facilitate future dissemination and funding initiatives... This study lays the groundwork for future research that explores how online news media could be better incorporated into dissemination processes and knowledge translation strategies." You also cite the 2018 Brownson article but don't discuss it or any related articles within this section. More consideration within the Discussion is warranted, as it seems to be a logical "next step" for this type of work.

Thank you for the suggestion. We have incorporated additional information from scientific communication literature regarding scientists communicating with journalists. We have included Brownson (2018) in the text and by name in the Conclusion. We have also worked to better address dissemination processes and knowledge translation strategies in the Conclusion.

pp. 2–3, 14–16 

We greatly appreciate your further review of our manuscript and hope that the revisions as described above and in the revised paper meet your expectations. Please do not hesitate to contact me with any questions or additional requests. 

Best regards,

Dr. Laura Moorhead

Assistant Professor, Journalism

---

## [Decision Letter · Decision Letter 1]

4 Dec 2020

PONE-D-20-15200R1

What cancer research makes the news?

A quantitative analysis of online news stories that mention cancer studies

PLOS ONE

Dear Dr. Moorhead,

Thank you for submitting your manuscript to PLOS ONE. After careful consideration, we feel that it has merit but does not fully meet PLOS ONE’s publication criteria as it currently stands. Therefore, we invite you to submit a revised version of the manuscript that addresses the points raised during the review process.

Specifically:

1) Please address the issues raised by the reviewers on data collection, statistical method and visual presentation; 

2) Please add a data availability statement to the manuscript (see https://journals.plos.org/plosone/s/data-availability) and make sure that the original data are accessible;

3) Please make sure that all the necessary supplementary information is provided and correctly referenced in the manuscript (please carefully read https://journals.plos.org/plosone/s/submission-guidelines and https://journals.plos.org/plosone/s/supporting-information).

We look forward to receiving your revised manuscript.

Kind regards,

Cindy Sing Bik Ngai

Academic Editor

PLOS ONE

Reviewers' comments:

Reviewer's Responses to Questions

**Comments to the Author**

1. If the authors have adequately addressed your comments raised in a previous round of review and you feel that this manuscript is now acceptable for publication, you may indicate that here to bypass the “Comments to the Author” section, enter your conflict of interest statement in the “Confidential to Editor” section, and submit your "Accept" recommendation.

Reviewer #1: (No Response)

Reviewer #2: All comments have been addressed

Reviewer #3: All comments have been addressed

2. Is the manuscript technically sound, and do the data support the conclusions?

Reviewer #1: No

Reviewer #2: Yes

Reviewer #3: Yes

3. Has the statistical analysis been performed appropriately and rigorously? 

Reviewer #1: Yes

Reviewer #2: I Don't Know

Reviewer #3: Yes

4. Have the authors made all data underlying the findings in their manuscript fully available?

Reviewer #1: Yes

Reviewer #2: Yes

Reviewer #3: Yes

5. Is the manuscript presented in an intelligible fashion and written in standard English?

Reviewer #1: Yes

Reviewer #2: Yes

Reviewer #3: Yes

6. Review Comments to the Author

Reviewer #1: I'd like to thank the authors for the efforts done on addressing my comments. However, I still have strong concerns with the conclusions the authors provide, being, that US-funded cancer research is barely mentioned in the news media. I do not think the authors really show this, but that based on a very sensitive and prone to errors method, which is that of using hyperlinks to papers (a rare way of reporting research findings in news media) they are not able to identify many mentions to US-funded research. I think this is very important distinction, as it places the problem on the fact that we do not have tools which are appropriate to do what the authors aim at.

The authors work with very low numbers, and therefore I believe that the data could be further explored in order to describe, 1) how is Altmetric.com retrieving the mentions?, and 2) are these findings a reflection of what is happening (rare mentions to US-funded cancer research, or a limitation of the tool the authors are using?

For the first issue, simply going into the actual news mentioning the papers and looking at them (at least a sample, but the size of the data is so small, that it is manageable) to then report how the news are linking to research and the context in which it is made would be enough. For the second, a potential idea would be to search in a news media database (the authors already mention two) for one of the more common cancers in a specific outlet (e.g., NY Times or any other), see how much they retrieve and see how much deals with research to find how they report research findings. If you find consistent evidence on them reporting in the way Altmetric.com identifies scientific literature, then you have some evidence on the robustness of your findings, even if it is anecdotal.

Reviewer #2: - I'm still not convinced the chi-squared is worth keeping; consider removing. It adds very little, if anything, to the work and without examining standardized residuals it's hard to know what a significant chi-squared even means in this context. If the authors decide to keep it, please make sure the numbers in the sentences are clear and correct (what are the 204 and 240?) and the assumptions for chi-squared are met.

- Regarding Figure 1, the ratios are a good idea, although the Figure then loses the impact about how serious each cancer is in terms of causing death. I'm including a version of the graph I sent in R in the previous review in Excel with the following adjustments from what was included in the paper:

(1) flip the coordinates so that the labels are easier to read

(2) order the bars by height

(3) add some way of knowing how serious each cancer is (either a second bar showing deaths/each cancer or some kind of labeling)

- I can't figure out where the code is for this paper from the zenodo link that was sent; what is saved there appears to be mostly stuff from the prior paper?

Reviewer #3: (No Response)

7. PLOS authors have the option to publish the peer review history of their article (what does this mean?). If published, this will include your full peer review and any attached files.

Reviewer #1: **Yes: **Nicolas Robinson-Garcia

Reviewer #2: **Yes: **Jenine K Harris

Reviewer #3: No

---

## [Author Response · Author response to Decision Letter 1]

3 Feb 2021

Dear editors,

Again, we thank you and the reviewers for your feedback and suggestions regarding our manuscript entitled, "What cancer research makes the news? A quantitative analysis of online news stories that mention cancer studies" (Manuscript PONE-D-20-15200) by Moorhead, Krakow and Maggio. We are pleased to resubmit our manuscript with revisions based on this feedback. In attached response to reviewers, we provide a description of how we addressed each reviewer comment. These revisions are also noted within the document using track changes.

We greatly appreciate your further review of our manuscript and hope that the revisions as described above and in the revised paper meet your expectations. Please do not hesitate to contact me with any questions or additional requests. 

Best regards,

Dr. Laura Moorhead

Assistant Professor, Journalism

---

## [Editor Report · Decision Letter 2]

10 Feb 2021

What cancer research makes the news?

A quantitative analysis of online news stories that mention cancer studies

PONE-D-20-15200R2

Dear Dr. Moorhead,

We’re pleased to inform you that your manuscript has been judged scientifically suitable for publication and will be formally accepted for publication once it meets all outstanding technical requirements.

Kind regards,

Cindy Sing Bik Ngai

Academic Editor

PLOS ONE
---

## [Editor Report · Acceptance letter]

16 Feb 2021

PONE-D-20-15200R2 

What cancer research makes the news? A quantitative analysis of online news stories that mention cancer studies 

Dear Dr. Moorhead:

I'm pleased to inform you that your manuscript has been deemed suitable for publication in PLOS ONE. Congratulations! Your manuscript is now with our production department. 

Kind regards, 

on behalf of

Dr. Cindy Sing Bik Ngai 

Academic Editor

PLOS ONE